rsos.royalsocietypublishing.org

**Cite this article:** Özbakır Y, Jonáš A, Kiraz A, Erkey C. 2018 A new type of microphotoreactor with integrated optofluidic waveguide based on solid-air nanoporous aerogels. *R. Soc. open sci.* **5**: 180802.

photochemistry/microsystems/nanotechnology

microphotoreactors, photochemistry, aerogels, nanostructured materials, optofluidic waveguides

**Author for correspondence:**
Can Erkey
e-mail: cerkey@ku.edu.tr

# A new type of microphotoreactor with integrated optofluidic waveguide based on solid-air nanoporous aerogels

Yaprak Özbakır[1], Alexandr Jonáš[4], Alper Kiraz[2,3]
and Can Erkey[1]

[1]Department of Chemical and Biological Engineering, [2]Department of Physics, and [3]Department of Electrical and Electronics Engineering, Koc University, 34450 Sarıyer, Istanbul, Turkey
[4]The Czech Academy of Sciences, Institute of Scientific Instruments, Královopolská 147, 612 64 Brno, Czech Republic

  AJ, 0000-0002-3555-6901; AK, 0000-0001-7977-1286;
CE, 0000-0001-6539-7748

In this study, we developed a new type of microphotoreactor based on an optofluidic waveguide with aqueous liquid core fabricated inside a nanoporous aerogel. To this end, we synthesized a hydrophobic silica aerogel monolith with a density of $0.22\,\mathrm{g\,cm^{-3}}$ and a low refractive index of 1.06 that—from the optical point of view—effectively behaves like solid air. Subsequently, we drilled an L-shaped channel within the monolith that confined both the aqueous core liquid and the guided light, the latter property arising due to total internal reflection of light from the liquid–aerogel interface. We characterized the efficiency of light guiding in liquid-filled channel and—using the light delivered by waveguiding—we carried out photochemical reactions in the channel filled with aqueous solutions of methylene blue dye. We demonstrated that methylene blue could be efficiently degraded in the optofluidic photoreactor, with conversion increasing with increasing power of the incident light. The presented optofluidic microphotoreactor represents a versatile platform employing light guiding concept of conventional optical fibres for performing photochemical reactions.

## 1. Introduction

Optofluidics is an emerging research field that combines optics and photonics with microfluidics in the same platform and

exploits the synergy between the unique features of both disciplines for a wide range of applications including biological sensing [1,2], chemical analysis [1], imaging [3–5], detection [2,6–8] and energy conversion [9–14]. Optofluidics allows for simultaneous delivery and control of light and fluids with microscopic precision. Since the light and the fluid share the same space within an optofluidic system, photons can be efficiently transferred to the fluid. Conversely, adjustment of optical properties of the fluid directly affects the photon propagation path.

Optofluidic microphotoreactors, in which light-activated photochemical reactions are carried out in solutions confined within microfluidic channels, are attracting increasing attention for a wide variety of applications ranging from photochemical synthesis to waste water treatment [15–18]. The maximized overlap between light and solutions residing in small reaction volumes enables highly efficient photochemistry at incident optical powers smaller than those used in conventional systems, with improved spatial homogeneity of irradiation and better light penetration through the reaction volume compared to the conventional large-scale photoreactors [12,17,19,20].

A variety of optofluidic microphotoreactors have been described in the literature, which generally consist of microchannels fabricated in a solid substrate covered by a transparent plate sealing the channels from the top. Microchannels have been directly fabricated in glass, polymer and ceramic substrates using various methods such as photolithography, dry/wet etching and micro/nano imprinting [21–23]. The above described microphotoreactors are designed to function exclusively with external irradiation delivered through the transparent top wall of the channel. Therefore, the thicknesses of the transparent plate, the depth of the channel walls and the geometry of the microphotoreactor have to be carefully adjusted to minimize the absorption and reflection losses. However, despite optimizing the geometry, light access to confined spots in the reaction chamber might still be problematic [17,24–26].

In order to fully harness the potential of optofluidic microphotoreactors for carrying out light-driven chemical processes, a more efficient management of light distribution within the reaction volume is highly desirable. Interaction between reactants and light can be dramatically improved by enabling light propagation directly within the reactor, using the concept of light guiding. Integration of waveguides into microphotoreactors has been attracting increasing attention in recent years to overcome the limitations of conventional optofluidic microphotoreactors. A variety of optofluidic microphotoreactors featuring light guides have been recently described in the literature for carrying out both photocatalytic and photolytic reactions. One type of such a photoreactor consists of a bundle of photocatalyst-coated optical fibres, around which the reactant solution flows. Optical fibres represent archetypal optical waveguides that can confine and deliver light at high intensities over long distances. They consist of a solid core surrounded by a solid cladding and achieve light guiding by total internal reflection (TIR) through the core resulting from the refractive index contrast between the core and the cladding (in particular, $n_{core} > n_{cladding}$). Due to this index contrast, the light incident upon the interface between the fibre core and cladding from the core side at an angle greater than the critical angle is repeatedly reflected down the fibre length [4]. While most of the light intensity propagates along the fibre, a small evanescent fraction refracts, penetrates through the cladding and excites the photocatalyst layer at the outer surface of the fibre, thus creating electron–hole pairs that subsequently initiate photocatalytic reactions. This concept was first used by Hofstadler *et al.* [27] who designed a photoreactor in a glass tube containing a bundle of photocatalyst-coated fibres and an inlet/outlet for reactants and products. The photoreactor was successfully employed for photodegradation of 4-chlorophenol in water. Peill & Hoffmann [28,29] developed a similar fibre-based photoreactor system and studied photocatalytic degradation of pentachlorophenol, 4-chlorophenol, dichloroacetate and oxalate in water. Wu *et al.* [30] also used a similar photoreactor system that consisted of nearly 120 $Cu/TiO_2$-coated fibres serving to transmit and uniformly distribute light inside the reactor for $CO_2$ reduction. Nguyen & Wu [31] used the same optical-fibre photoreactor to reduce $CO_2$ with $H_2O$ to hydrocarbons over $TiO_2/SiO_2$ mixed oxide-based photocatalyst. However, such evanescently coupled photoreactors are not efficient for non-catalytic photolytic reactions that occur in the bulk fluid in the microchannel rather than at the interface between the fluid and the walls of the microchannel where the catalyst usually resides.

Another kind of optofluidic microphotoreactor employs photonic crystal fibres with a hollow core (HC-PCF) surrounded by a cladding structure that consists of a periodic array of sub-micrometre-sized air-filled openings running along the entire length of the fibre [32]. Reactants can be introduced into the hollow core, in which the light is confined by photonic band gap formed due to the cladding holes. The reactants reside within the waveguide and almost completely overlap with the propagating light, resulting in excellent interaction efficiency. Chen *et al.* [33] used a liquid-filled HC-PCF with

'kagome' cladding (lattice of thin silica webs forming equilateral triangles and regular hexagons arranged so that each hexagon is surrounded by triangles) as an optofluidic microphotoreactor combined with absorption spectroscopy measurements for monitoring the progress of photoconversion. They demonstrated highly efficient photolysis of vitamin $B_{12}$ (cyanocobalamin, CNCbl) in aqueous solution using a low-power laser light source. The hollow core of the fibre was filled with the reactant solution and light from a diode laser operating at the wavelength of 488 nm was coupled into the core and guided through the solution. The water-filled fibre transmitted light over the whole visible spectral range between 400 and 750 nm, with wavelength-dependent propagation loss ranging from approximately 10 to 30 dB m$^{-1}$. A similar experimental set-up was used by Williams *et al*. [19] to investigate the kinetics of photochemical and thermal isomerization of azobenzene-based dyes, Disperse Red 1 (N-ethyl-N-(2-hydroxyethyl)-4-(4-nitrophenylazo)aniline) and Disperse Orange (1,4-anilino-4′-nitroazobenzene). Unterkofler *et al*. [34] used HC-PCF as a microphotoreactor coupled to a high-resolution mass spectrometer. The microphotoreactor was operated in the continuous-flow configuration and photolysis of CNCbl was investigated. The same HC-PCF-based optofluidic microphotoreactor was then used for photoactivation of a drug, di-nuclear ruthenium complex $[\{(\eta^6 - \mathrm{indan})\mathrm{RuCl}\}_2(\mu - 2,3 - \mathrm{dpp})](\mathrm{PF}_6)_2$ [35]. The products of photochemical reactions of the drug could be directly detected and analysed by mass spectroscopy. This combined optofluidic device proved very useful for rapid analysis of photoactivatable drugs on a timescale much shorter than that of the standard UV–Vis or NMR spectroscopy. However, despite their successful applications, HC-PCF-based microphotoreactors exhibit several drawbacks and limitations. Due to the small sizes of fibre hollow cores, only a low volume of reactants can be treated in HC-PCF photoreactors. Since the glass structure of the fibre is impermeable for gases, gaseous reactants and products of photochemical reactions cannot be readily exchanged between the reactor and its surroundings. Similarly, air bubbles formed in the liquid solutions confined within the fibre core cannot be easily removed, which results in high optical loss from scattering and eventually causes failure of the waveguide.

It has been recently demonstrated that aerogels can be used for efficient optical waveguiding in liquids [36–40]. Aerogels are highly porous nanostructured solids consisting of an interconnected open network of loosely packed, bonded particles separated by air gaps, and they feature very low refractive index and high specific surface area [36,40]. These properties, particularly the low refractive index of approximately 1.05, for which the aerogels are often referred to as 'solid air', make aerogels a remarkable solid-cladding material for optofluidic waveguides that can be used without any additional modification, since almost all liquids have refractive indices exceeding that of aerogels. Optofluidic waveguides can be constructed by opening channels inside monolithic aerogel blocks. These channels are surrounded by a wall made of interconnected particles that are around 40–80 nm in size, with pockets of air in between them that constitute pores with typical sizes less than 100 nm. The air confined in the aerogel pockets is then responsible for guiding light by TIR in a liquid-filled channel. Following appropriate chemical treatment that makes the channel surface compatible either with polar or with non-polar liquids, the channels can be filled with a suitable reactant solution that also serves as the waveguide core liquid. In such an approach, the channels fabricated within the aerogel simultaneously confine the reaction medium and also serve as the waveguide cladding, whereas the reaction volume resides within the liquid core of the waveguide. The propagation of non-lossy optical modes guided in the liquid by TIR from the channel walls requires the cladding material to have a low absorption coefficient at the working light wavelength and a lower refractive index than that of the core liquid [37,40]. These conditions can be easily met over the whole visible and near-UV spectral regions by, for example, silica aerogels. In addition, the surface of aerogels can be chemically modified; therefore, there is no restriction on the type of liquid or aerogel that can be used, as long as the liquid can be confined inside the aerogel block without penetrating its porous network [40]. Therefore, it should be possible to use such aerogel-based waveguides in microphotoreactors.

In this study, we demonstrate a new type of aerogel-based microphotoreactor with an integrated optofluidic waveguide. This optofluidic microphotoreactor consists of a single liquid-filled channel fabricated inside a monolithic, hydrophobic silica aerogel, within which photochemical reactions are carried out. Due to the contrast of refractive indices between the aerogel and the aqueous working liquid, optofluidic waveguides based on TIR are naturally formed to deliver light to the liquid reaction medium inside the microchannel, as shown schematically in figure 1. We demonstrate that this configuration provides an excellent overlap between the guided light and the liquid in the channel for efficient photochemical activation. We show that our microphotoreactor is well suited for photochemical degradation of a model organic compound—methylene blue dye—and we characterize

rsos.royalsocietypublishing.org    R. Soc. open sci. **5**: 180802

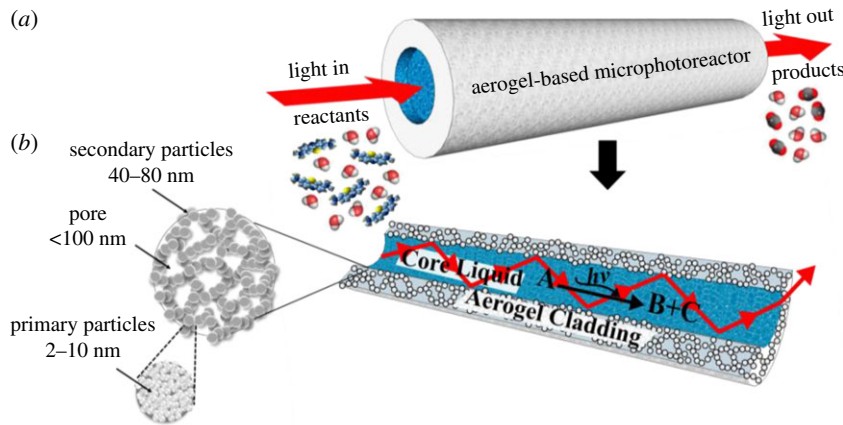

**Figure 1.** (*a*) Aerogel-based microphotoreactor with an integrated optofluidic waveguide (*b*) Cross-sectional view of the microphotoreactor/waveguide system illustrating light guiding in the liquid by TIR.

the efficiency of the dye photoconversion as a function of the incident light power. Side-by-side comparison with a microphotoreactor based on conventional polymer tubing then provides direct evidence that the light-guiding effect is indeed essential for the optimal performance of the microphotoreactor.

# 2. Material and methods

## 2.1. Materials

Tetraethylorthosilicate (TEOS) (98%), ethanol (≥98%) and ammonium hydroxide (NH$_4$OH) (2.0 M in ethanol) were purchased from Sigma Aldrich. Hydrochloric acid (HCl) (37% purity) was obtained from Riedel-de Haen and hexamethyldisilazane (HMDS) (≥98%) was purchased from Merck. Carbon dioxide (99.998%) was purchased from Messer Aligaz. All chemicals were used as received.

## 2.2. Synthesis of hydrophobic silica aerogels

Silica aerogels were prepared using the sol-gel protocol described in detail in our recent study [40]. Briefly, TEOS was used as the silica precursor and the synthesis was carried out in water/ethanol reaction medium, using HCl as the acid catalyst in hydrolysis reactions and NH$_4$OH as the base catalyst for condensation reactions. The molar ratio of TEOS : ethanol : water : HCl : NH$_4$OH was 1 : 4 : 3 : 0.0023 : 0.009. Upon transfer to a rectangular plexiglass mould, the solution of dispersed silica nanoparticles was transformed into a continuous three-dimensional network of silica nanobeads and, subsequently, the resulting structure was aged in an ageing solution (40 v/v % TEOS, 10 v/v % water, 50 v/v % ethanol) in an oven at 50°C for 24 h and then at room temperature for an additional 3 days. The alcogels were purified in fresh ethanol for 3 days and finally dried with supercritical CO$_2$. The resulting monolithic silica aerogels were hydrophilic due to the presence of polar hydroxyl groups within their porous framework that promote high capillary stress and water adsorption once they are in contact with water. Therefore, the aerogel samples were made hydrophobic by replacing the hydrophilic hydroxyl surface groups with hydrophobic methyl groups via HMDS vapour treatment. At the end of this procedure, monolithic and crack-free hydrophobic silica aerogel was obtained.

## 2.3. Characterization of silica aerogels

As described previously in [40], average pore size, pore size distribution and surface area of the samples were determined using nitrogen adsorption–desorption measurements carried out at 77 K and relative pressure ($P/P_0$) ranging from $10^{-7}$ to 0.999. Total pore volume of the sample was determined by converting the adsorbed N$_2$ volume at STP to liquid N$_2$ volume at 77 K. Bulk density of monolithic aerogel samples was determined by dividing their mass by their final volume that was obtained by measuring their physical dimensions using a caliper. Hydrophobicity of HMDS-treated aerogels was quantified by static contact angle measurements at room temperature (23 ± 1°C) based on direct

rsos.royalsocietypublishing.org     R. Soc. open sci. **5**: 180802

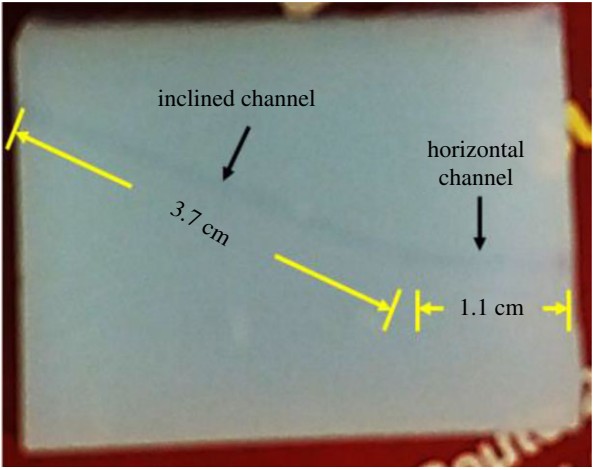

**Figure 2.** Side view of an aerogel monolith with the fabricated inclined L-shaped channel.

imaging of water droplets deposited on the surface of the sample. Deionized, triple distilled water (10 µl) was used in the contact angle measurements. In addition, wetting properties of the internal channel walls were characterized by cutting the aerogel sample along the channel axis and placing a water droplet directly on the inner channel surface. Fourier transform infrared spectroscopy-attenuated total reflectance (FTIR-ATR) spectra of hydrophilic and hydrophobic aerogel samples were recorded using a Thermo Scientific Smart iTR spectrometer to investigate the chemical composition of the samples before and after the HMDS treatment. Refractive index of aerogel monoliths was determined by measuring the angle of refraction of a laser beam from the aerogel block placed on a goniometer stage, as described in [40].

## 2.4. Fabrication of channels in aerogel monoliths

The technique demonstrated in [40] was used for the channel fabrication. An L-shaped channel consisting of a horizontal section intersecting with an inclined section was formed in a hydrophobic aerogel monolith by two-step manual drilling, using a 2.1 mm diameter drill bit. First, a straight horizontal channel with a length of 1.1 cm was opened from one of the two narrow side faces of the aerogel block. Subsequently, another inclined channel of 3.7 cm length was delicately drilled starting from the opposite side face of the block with an angle of about 30° with respect to the horizontal, up to the end of the initially created horizontal channel. Side view of the resulting L-shaped channel is provided in figure 2.

## 2.5. Experimental set-up for carrying out photochemical reactions

Figure 3 shows a schematic diagram of the experimental set-up that was employed for studying model photochemical reaction—degradation of methylene blue (MB) dye in an aqueous solution contained inside the microchannel fabricated within a hydrophobic silica aerogel. Hydrophobic aerogel block with the inclined L-shaped channel was mounted on an adjustable metal holder. A small piece of plastic tubing was glued to the open end of the horizontal channel section by epoxy and the other end of the tubing was connected to the side port of a Union Tee adapter. Central port of the Union Tee adapter was then connected to a syringe and the remaining port of the Union Tee adapter was used to insert a solarization-resistant multimode optical fibre (Thorlabs; UM22-300, NA = 0.22) into the aerogel channel. This fibre served for delivering the photoactivation light to the reaction volume and its end facet was held at a fixed position at the channel entrance. The channel was filled with the aqueous MB solution using the syringe. Due to the hydrophobic walls of the channel, MB solution could be confined within the channel without penetrating into the porous network of the aerogel. In order to initiate light-induced degradation of MB, we used a laser beam from a femtosecond-pulsed, tunable Ti:Sapphire laser light source (Coherent Chameleon) with a maximal output power of 4 W at the emission wavelength of 800 nm, featuring a frequency-doubling unit that further expanded the spectral tuning range of the laser. The laser beam was coupled into the free end of the multimode optical fibre with the aid of an objective lens. For experiments with MB photolysis, the operating

rsos.royalsocietypublishing.org    R. Soc. open sci. **5**: 180802

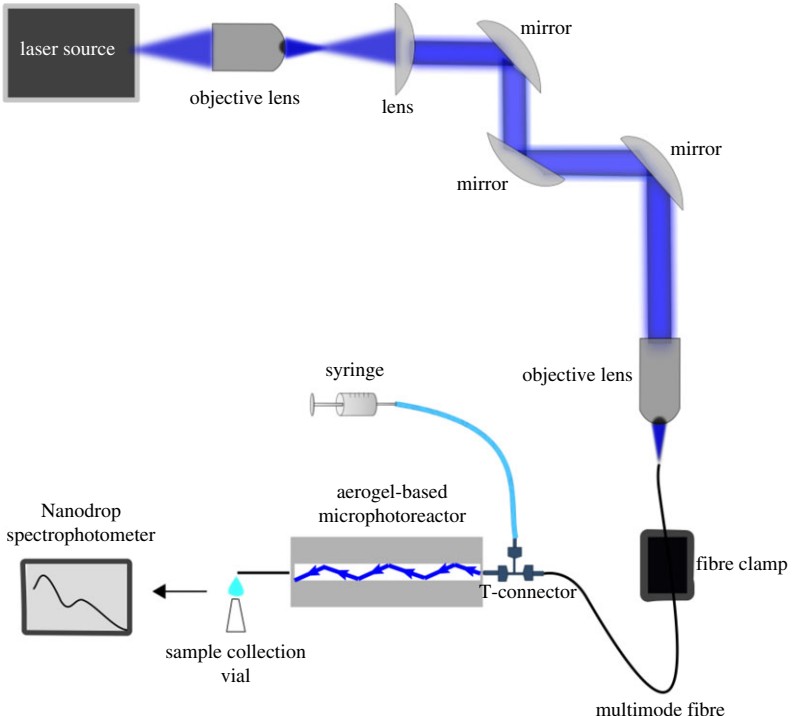

**Figure 3.** Schematics of experimental set-up used in the characterization of light guiding in aerogel-based optofluidic microphotoreactors and carrying out photochemical reactions in these microphotoreactors. See text for detailed description of individual components.

wavelength of the laser was tuned to 388 nm. The input power coupled into the optofluidic waveguide could be adjusted either from the laser source or by using neutral density filters. The incident power values reported in this work were measured in front of the objective lens that served for coupling the light into the optical fibre. Due to the losses introduced by the beam focusing optics and imperfect coupling of light into the fibre, about 17% of the incident light power actually reached the sample. The intensity of light transmitted through the waveguide was visually monitored and measured at the outlet of the channel by a power meter. In order to monitor the progress of photodegradation of MB, samples with a volume of 10 µl were collected at various times from the end of the channel by a micropipette and subsequently analysed by Nanodrop ND-1000 Spectrophotometer for absorbance-based quantification of MB concentration.

# 3. Results and discussion

## 3.1. Properties of synthesized aerogel samples

Synthesis of aerogel monoliths with high mechanical strength and sufficiently low refractive index is a critical step for fabricating our photoreactors [40]. We met these requirements by ageing of intermediate alcogels in a solution containing the silica precursor (see 'Material and methods' for additional details on the procedure). Reactions between partially hydrolysed TEOS in the ageing solution and silanol groups on the silica surface resulted in the formation of additional siloxane bonds and increased the solid content of the porous aerogel. These additional condensation reactions enhanced the mechanical strength and stiffness of the aerogel, most likely by increasing the size of the connecting silica necks within the solid network. After ageing, the resulting sample had a density of $0.22 \, \text{g cm}^{-3}$. The critical angle of incidence for a laser beam incident upon the aerogel–air interface from the aerogel side was measured to be 71°, which translated into the aerogel refractive index of 1.06 at 632.8 nm, much lower than the refractive index of an aqueous solution. Therefore, aged silica aerogel sample with a density higher than the silica aerogel prepared using the conventional two-step sol-gel method without any additional ageing is still suitable for TIR-based optofluidic waveguides without the use of any additional optical coatings of the channel inner walls.

rsos.royalsocietypublishing.org    R. Soc. open sci. 5: 180802

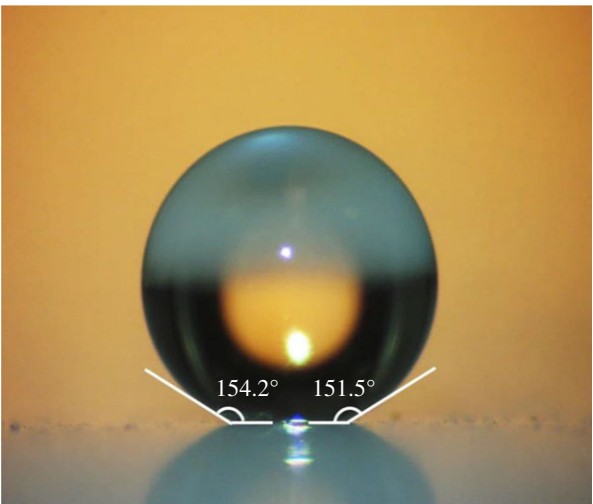

**Figure 4.** Water droplet resting on the surface of HMDS-treated silica aerogel monolith.

The reaction of surface silanol groups with HMDS vapour resulted in aerogel monoliths impermeable to aqueous solutions. As shown in figure 4, average contact angles were found to be greater than $150°$, showing that the samples are, in fact, superhydrophobic. Furthermore, their superhydrophobic nature was preserved during the experiments over several weeks, as water did not penetrate into the aerogel pores and did not crack the aerogel blocks in the course of the experiments. In addition to the contact angle measurements, both HMDS-treated and untreated samples were studied by infrared spectroscopy to investigate their chemical composition. FTIR-ATR measurements were carried out for samples that were crushed into a powder. In order to acquire the spectrum of the inner surface of the channel fabricated in an aerogel monolith, the channel was broken along its axial direction and the surface of the channel walls was gently ground with the aid of a spatula tip. FTIR-ATR measurement was then performed with thus collected powder. Figure 5 presents the FTIR-ATR spectra of originally synthesized hydrophilic aerogels and HMDS-treated hydrophobic aerogels. The absorption bands observed in the spectra were interpreted on the basis of the data available in the literature [41–44]. In particular, in the spectrum of hydrophilic silica aerogel (figure 5a), broad bands centred around 960 and 3400 $cm^{-1}$ represent stretching of the surface silanol (Si–OH) groups. A strong and broad band at approximately 1090 $cm^{-1}$ and a shoulder at approximately 1200 $cm^{-1}$ correspond to Si–O–Si asymmetric stretching vibrations. Intense peak in that spectral region indicates the presence of a dense silica network. A peak at approximately 800 $cm^{-1}$ is then due to symmetric stretching vibrations of Si–O–Si. The comparison of the spectra of untreated (figure 5a) and HMDS-treated (figure 5b) samples clearly indicates that the aerogels have been modified, since the intensities of the broad Si–OH band around 3400 $cm^{-1}$ and the other Si–OH peak around 960 $cm^{-1}$ are reduced upon treatment [43,44]. In addition, the presence of a sharp peak around 2900 $cm^{-1}$ corresponding to $CH_3$ symmetric and asymmetric vibrations and a peak at 837 $cm^{-1}$ corresponding to Si–C stretching vibrations indicates that HMDS indeed reacted with silanol groups on the backbone of the treated samples.

The pore properties of the resulting HMDS-treated aerogels were determined by using nitrogen physisorption. The specific pore volume, specific surface area and average pore size of the sample were measured as 2.5 $cm^3\,g^{-1}$, 549 $m^2\,g^{-1}$ and 17 nm, respectively. Samples prepared using the standard solvent exchange process without additional ageing have commonly reported specific pore volume of 4.0 $cm^3\,g^{-1}$, specific surface area of 1000 $m^2\,g^{-1}$ and an average pore size of 20 nm [45]. The slightly lower specific surface area, specific pore volume and pore size of modified aerogels with higher mechanical strength can be most likely attributed to the ageing process in TEOS solution that probably blocks or closes some of the pores.

## 3.2. Light-guiding in aerogel-based microphotoreactors

Light-guiding experiments with aerogel-based optofluidic microphotoreactors/waveguides were performed using the experimental set-up shown in figure 3. In order to visualize the distribution of light intensity within the waveguide, the horizontal part of the channel fabricated within the aerogel

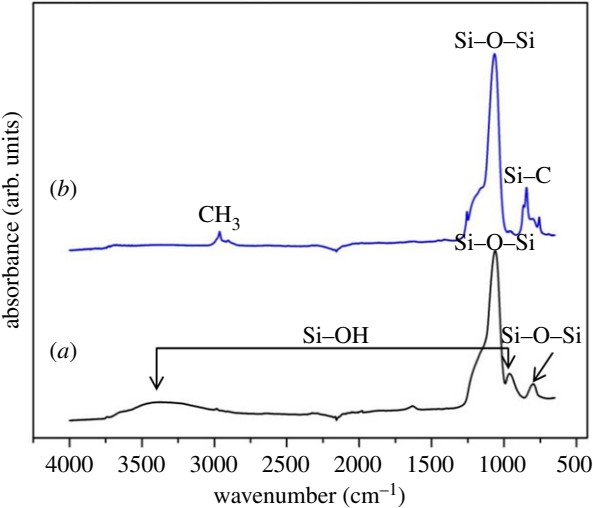

**Figure 5.** FTIR-ATR spectra of (*a*) an untreated hydrophilic silica aerogel and (*b*) the surface of the channel fabricated in a HMDS-treated hydrophobic silica aerogel. See text for explanation of individual spectral peaks.

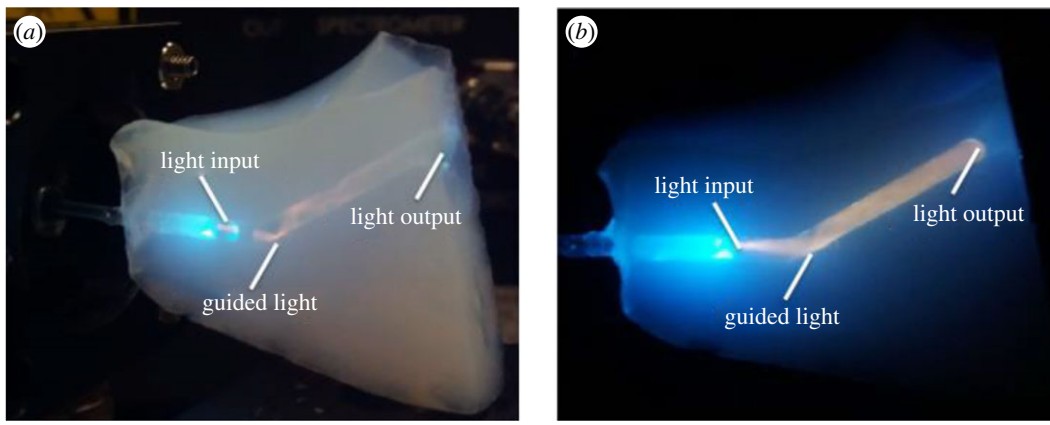

**Figure 6.** (*a*) Propagation of light delivered by an optical fibre into a channel fabricated in an aerogel monolith and filled with aqueous MB solution under ambient illumination (side view). (*b*) Propagation of light delivered by an optical fibre into in the same channel as shown in part (*a*) without ambient illumination (side view). The guided light fills the whole volume of the channel.

monolith (see figure 2 for illustration of the channel geometry) was illuminated by laser light tuned to 488 nm, coupled into the channel through the multimode optical fibre. When the light was coupled into an empty channel, the beam exiting from the fibre was transmitted straight along the $x$-axis (the direction of the horizontal channel section), without any light guiding into the inclined section of the channel, and intense scattering could be observed from the junction of the two sections of the channel. Subsequently, a multimode optofluidic waveguide was formed by filling the channel with an aqueous MB solution that has a higher refractive index than the aerogel. Owing to the prior hydrophobic treatment of the aerogel, the solution could be confined within the channel fabricated in the aerogel without penetrating into its porous network. When the light was coupled into the channel filled with the aqueous MB solution, the solution served both as the waveguide core liquid and the reaction medium and the light was guided along the full length of the channel including its inclined section, finally exiting from the opposite end of the channel (seen at the top of the right lateral face of the aerogel block shown in figure 6). Orange colour of the illuminated liquid-filled channel results from omnidirectional fluorescence emission of MB dye excited by the guided laser light with the wavelength of 488 nm. When the laser light rays propagating in the channel strike the channel wall at an angle greater than the critical angle, they are reflected back to the solution. The rays then strike the opposite wall of the channel and are again reflected back to the solution. As illustrated in the photographs of the solution-filled channel with coupled laser light shown in figure 6*a*,*b*, the laser light was totally internally reflected in the channel several times after emerging from the fibre, before leaving the waveguide. Consecutive reflections appear more faintly under the ambient illumination;

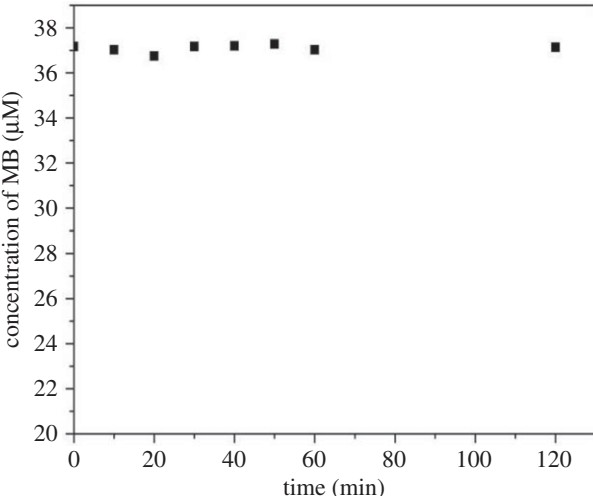

**Figure 7.** Time variation of the concentration of aqueous MB solution contained in an aerogel channel kept in the dark. The initial concentration of MB was 37.2 μM.

however, it is clearly visible in the dark that the entire length of the channel is filled with guided light, which indicates that the light is indeed confined within the liquid-core optofluidic waveguide.

The performance of optofluidic waveguides can be characterized in terms of the waveguide propagation losses that can in turn be determined from the measurement of the input optical power and the output power observed after the light has propagated a certain distance $z$ along the waveguide [36–38,40]. Our measurements with the same waveguide geometry reported recently in [40] have shown that the propagation losses of multimode aerogel waveguides with aqueous cores are below $-1.5$ dB cm$^{-1}$. This favourable value can be attributed to the high contrast of the refractive index between the aqueous core and 'solid air'-like cladding of these waveguides.

## 3.3. Photodegradation of organic compounds

MB was chosen as a model organic compound to evaluate the performance of our aerogel-based photoreactor with integrated optofluidic waveguide in photochemical reactions. An aqueous solution of MB with an initial concentration of 37.2 μM was loaded into an L-shaped aerogel microchannel. L-shaped microchannel was chosen instead of a straight microchannel in order to make sure that the sample of the MB solution taken from the exit port of the photoreactor was exposed only to the light guided along the channel by TIR and not to the light emitted directly from the input optical fibre. Consequently, light-induced degradation of MB at the exit port of the photoreactor occurred only due to waveguiding in the microchannel.

Since MB might be gradually adsorbed on the surface of the aerogel channel, the concentration of MB solution collected from the channel at various times after starting the experiment might be lower than the initial concentration of the solution due to adsorption rather than photolytic degradation. In order to find out if there is a decrease in MB concentration due to adsorption, aerogel channel filled with MB solution was first kept in the dark for a certain period of time. Samples were periodically collected from the end of the channel and the concentration of MB in these samples was measured using Nanodrop Spectrophotometer. Figure 7 shows that the measured concentration of MB solution kept in the dark remained nearly constant for 120 min, with a very little random variation, thus indicating that MB was not adsorbed at the channel surface. FTIR-ATR spectra acquired from the internal surface of the channel, which are provided in figure 8, then reveal that the chemical composition of the channel walls remained unchanged before and after MB photolysis took place in the channel. Therefore, they also confirm that MB was not adsorbed at the channel walls. Low affinity of MB for the aerogel surface may be attributed to the surface hydrophobicity, in combination with the fact that a very small amount of solid constitutes the actual liquid–aerogel interface.

Subsequently, activation laser light with the wavelength of 388 nm was coupled into the aerogel channel filled with MB solution with an initial concentration of 36 μM and propagated to the end of the channel by TIR-assisted waveguiding. Under these experimental conditions, MB could be gradually degraded along the full length of the channel by the guided light. The degree of

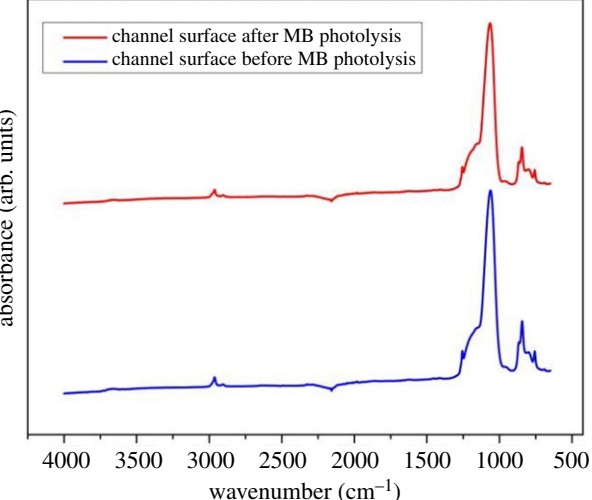

**Figure 8.** FTIR-ATR spectra of internal surfaces of channels fabricated in hydrophobic silica aerogel monoliths acquired before (blue) and after (red) MB photolysis took place in the channel. For the sake of clarity, the spectra have been vertically shifted with respect to each other.

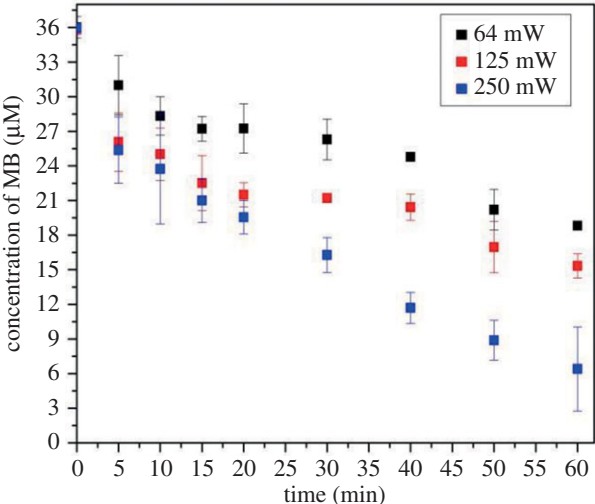

**Figure 9.** Time variation of the concentration of aqueous MB solution contained in an aerogel channel under illumination at 388 nm with varying power of the incident light. The initial concentration of MB was 36 μM. The incident power values were measured before coupling the light into the input optical fibre.

photodegradation then depended on the light exposure time and the incident light power. Figure 9 demonstrates that the concentration of MB samples collected from the channel end decreased with time under illumination with a constant power of the incident light. In particular, within 60 min, the concentration of MB decreased to 6.4 μM under the exposure with the highest incident light power of 250 mW, while it reduced to 15.3 μM with the incident light power of 125 mW and to 18.8 μM under the exposure with the incident light power of 64 mW. The photoconversion of MB dye compound was calculated as:

$$\text{Conversion (\%)} = \frac{C_0 - C_t}{C_0} \times 100. \tag{3.1}$$

Here, $C_0$ is the initial concentration of the dye and $C_t$ is the dye concentration at time $t$. As shown in figure 10, within the studied power range, the photoconversion of MB dye at various reaction times monotonically increased with increasing power of the incident light.

In our photoreactor geometry, the light is coupled to one end of the channel (figure 6) and the intensity $I$ of the activation light is highest at the channel entrance and gradually decreases towards

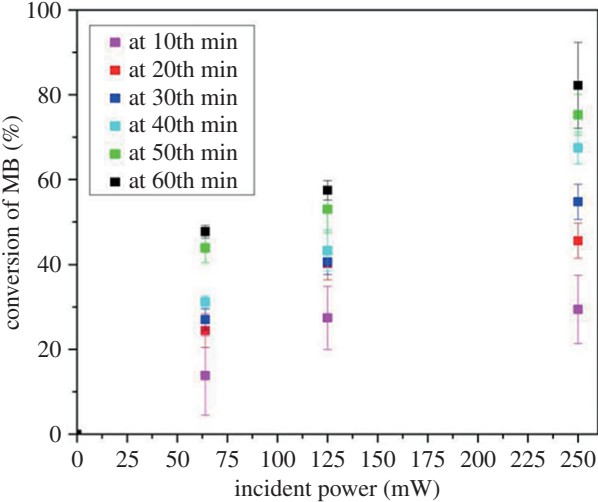

**Figure 10.** Variation of the photoconversion of aqueous MB solution illuminated by incident light at 388 nm with increasing power. The initial concentration of MB was 36 μM.

the other end of the channel due to absorption and scattering. Therefore, the photodegradation rate is highest at the entrance and decreases towards the end of the channel. Consequently, this leads to a concentration gradient of MB along the channel axis. As a result, MB diffuses from the high-concentration region near the end of the channel to the region with a lower concentration of MB near the channel entrance. The change of concentration $C$ of MB dye in the channel as a function of time and position is given by the solution of the following system of coupled integro-differential equations:

$$\frac{\partial C}{\partial t} = \frac{\partial}{\partial z}\left(D_{AB}\frac{\partial C}{\partial z}\right) - \Phi * I_{a} \tag{3.2}$$

and

$$I_{a} = \left[I_{o}\,e^{-2.303*\varepsilon*\int_{0}^{z} C(z)*dz}\right]*2.303*\varepsilon*C(z), \tag{3.3}$$

where $D_{AB}$ is the diffusion coefficient of MB in water (cm$^2$ s$^{-1}$), $I_{o}$ is the incident light intensity at the channel entrance (E s$^{-1}$ cm$^{-2}$), $I_{a}$ is the locally absorbed light energy per unit volume and time (E s$^{-1}$ L$^{-1}$), $\Phi$ is the quantum yield of MB, which is defined as the number of moles of MB decomposed per mole of light photons absorbed by MB, $\varepsilon$ is the molar absorption coefficient of MB in aqueous solution (M$^{-1}$ cm$^{-1}$), and $z$ is the path length of the activation light measured along the channel axis (cm). Numerical solution of coupled equations (3.2) and (3.3) with boundary conditions applicable to our system indicates that the concentration of MB at the channel exit changes as a function of the incident light power in a complex manner.

In order to demonstrate the vital importance of light guiding for the functionality of our photoreactor system, we carried out control experiments, in which the aerogel block with liquid-filled channel was replaced by a piece of conventional plastic tubing. In particular, the used tubing had an outer diameter of 4 mm and an inner diameter of 2 mm and its L-shaped geometry and dimensions were almost identical to those of the channel fabricated in the aerogel monolith. In order to maintain constant shape during the course of an experiment, the tubing was mounted on a metal plate using clamps (see figure 11 for illustration). Subsequently, the tubing was connected to the Union Tee adapter in the same way as in the experiments conducted with the aerogel (see §2.5). Prior to the actual photodegradation experiments, light propagation in the tubing filled with MB solution was investigated. The light at 488 nm was coupled into the tubing by the optical fibre, using the same optical set-up as used previously in the aerogel experiments (see §3.2). A photograph of the tubing filled with MB solution and 488 nm laser light coupled into it is shown in figure 11. As expected, the laser light delivered by the fibre cannot propagate in the liquid-filled tubing by TIR, since the refractive index of the tubing is higher than the refractive index of water. Thus, a large fraction of the incident light coupled into the horizontal section of the tubing is scattered at the tubing elbow. Only a small fraction of the incident light can reach the end of the tubing, exploiting TIR at the interface between the tubing and ambient air. Subsequently, the tubing filled with fresh MB solution was kept in the dark and samples were collected from the end of the tubing at various times up to

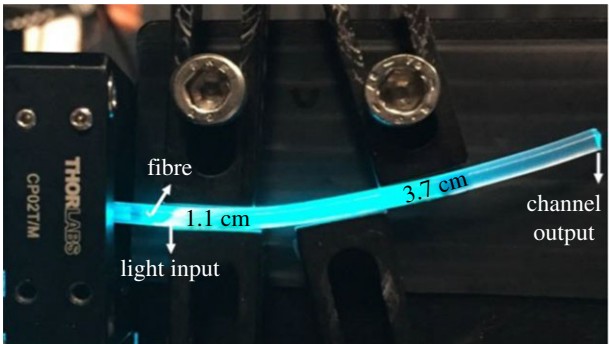

rsos.royalsocietypublishing.org    R. Soc. open sci. **5**: 180802

**Figure 11.** Propagation of light delivered by an optical fibre into a plastic tubing filled with aqueous MB solution under ambient illumination (side view).

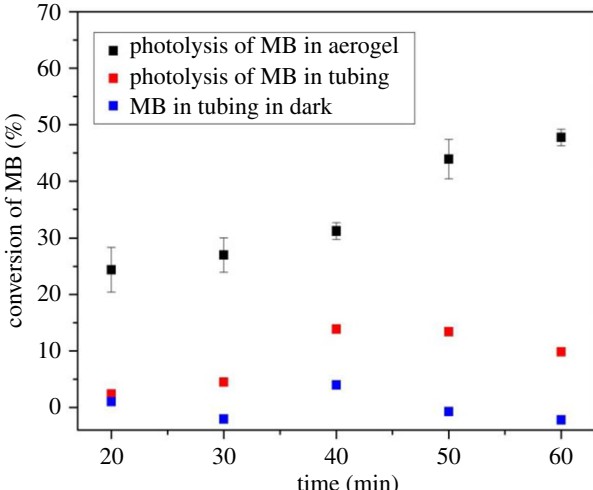

**Figure 12.** Time variation of the conversion of aqueous MB solution contained in a plastic tubing under dark conditions (blue) and under illumination at 388 nm with an incident light power of 64 mW (red). The initial concentration of MB was 30 μM. For comparison, time variation of the conversion of aqueous MB solution contained in an aerogel channel under illumination at 388 nm with an incident light power of 64 mW is also shown (black).

60 min. Figure 12 illustrates that in this experiment, MB concentration remained almost constant, with a slight variation that might be possibly attributed to interaction between MB and the surface of the inner walls of the tubing. After completing the control experiment under dark conditions, photolysis of MB contained in the tubing was carried out under illumination at 388 nm with an incident light power of 64 mW, measured before coupling the light into the input optical fibre. The power at the tubing output end was measured to be lower than approximately 8 μW. Figure 12 shows that in the photolysis experiment carried out in the tubing, the conversion of MB increased only slightly with time, eventually fluctuating around the maximum of approximately 10% after 60 min exposure (red squares). In contrast, the conversion of MB carried out in the aerogel-based waveguide (black squares) reached the maximum of 48% under the same experimental conditions (incident light power and exposure time). The comparison of the results obtained with aerogel-based and tubing-based photoreactors clearly shows that the light-guiding effect enabled by the contrast of refractive index between the aerogel and the aqueous reaction solution is essential for the high performance of the microphotoreactor.

The above results demonstrate that aerogel-based optofluidic waveguides can serve to build efficient optofluidic photoreactor systems for a wide variety of potential applications. Thanks to the integrated light waveguide, these photoreactors enable a large effective penetration depth of the incident light and a more uniform light distribution within the reaction volume, in comparison to the conventional bulk photoreactors. As the light is delivered to the reacting species in the liquid core by optical waveguiding, the reactor itself can be operated under dark ambient conditions. Furthermore, thanks to the interconnected open-pore architecture of the aerogel network forming the waveguide cladding, the

platform is also interesting for implementing membrane-like microreactors with selective molecular transport. In particular, while the core liquid is confined within the channel and cannot penetrate the porous aerogel cladding, the pores still allow for efficient two-way diffusive transport of gases between the core liquid and the reactor surroundings. Thus, oxygen required for the photolysis of organic compounds such as methylene blue can be delivered from the outside whereas the gaseous products of the photolysis such as $CO_2$ can be readily removed through the porous framework of the aerogel. Moreover, the selectivity of transport through the porous aerogel network can be tuned for specific applications by adjusting the parameters of sol-gel synthesis and post-synthesis treatment of the aerogels. Permeability of aerogels to gases also allows for the removal of air bubbles formed in the liquid channel, leaving behind a continuous, bubble-free light path. Air bubbles are detrimental for the performance of optofluidic waveguides since they hinder the light propagation through the channel and lead to high optical losses due to light scattering, eventually resulting in the failure of the waveguide. However, such bubbles cannot be removed from conventional microfluidic channels with gas-impermeable walls [37].

Finally, aerogel-based optofluidic microphotoreactors hold a great promise for carrying out photocatalytic reactions. The high porosity and high surface area of aerogels with interconnected open pore structure, combined with a fully tunable three-dimensional architecture, make aerogels very well suited for the deposition of a photocatalyst. Walls of channels fabricated in an aerogel monolith are porous and, thus, photocatalyst particles can be immobilized within these pores with a good adhesion to the solid network. With photocatalyst particles present in the aerogel channel wall, the guided light can be partially absorbed in the near-surface region of the wall. Subsequently, the reactants can be converted into desired products on the catalyst surface.

Even though we used our optofluidic photoreactors in the static regime with no liquid flow, it should be also possible to operate them in a continuous manner by delivering the reactant solution using a suitable pump. In order to increase the overall throughput of the photochemical/photocatalytic reaction, the aerogel-based photoreactors can be fabricated with multiple channels operating in parallel. Moreover, using processes such as preform removal or femtosecond laser ablation, uniform and extended microchannels with much smaller diameters can be created [36,38,39] expanding further the potential for large-scale parallelization that is needed for industrial applications of the presented platform.

# 4. Conclusion

We have demonstrated a novel type of microphotoreactor with integrated optofluidic waveguide that is formed by a liquid-filled channel fabricated in a monolithic aerogel block The unique optical properties of aerogels—particularly, their low refractive index—allow them to be used as the cladding material of TIR-based optofluidic waveguides with aqueous liquid cores that do not require any additional coatings. Post-synthesis modification of the prepared aerogel monoliths rendered the aerogel surface hydrophobic. Thus, aqueous solutions of methylene blue serving both as the waveguide core liquid and the reaction medium could be confined inside a channel embedded in the aerogel, without being adsorbed on the channel walls or without compromising the monolithic structure of the aerogel. We have directly visualized TIR-assisted light propagation along liquid-filled channels fabricated in aerogel blocks and verified that light could be efficiently guided even along paths with a curved geometry. Subsequently, we have successfully shown light-driven photolysis of methylene blue by the light guided along the full length of the photoreactor channel. For the studied range of incident light powers, quantitative analysis of the dependence of conversion of the dye on the incident power indicated a linear trend. Finally, we have demonstrated that the light-guiding effect enabled by the contrast of refractive index between the aerogel and the aqueous reaction solution is essential for the efficient performance of the microphotoreactor. Aerogel-based liquid-core optofluidic waveguides represent a straightforward way for guiding and controlled routing of light that is also fully compatible with carrying out photochemical and photocatalytic reactions in aqueous media. Thanks to the flexibility of the procedures used for aerogel synthesis and post-synthesis modifications, aerogel-based photoreactors hold the potential for practical applications in photochemical and photocatalytic synthesis or degradation of various organic and inorganic compounds.

Data accessibility. All data used in this research are included in figures.
Authors' contributions. Y.Ö. is a PhD student working on the preparation and characterization of aerogel-based microphotoreactors with integrated optofluidic waveguides; she carried out the experimental work. A.J., A.K. and C.E. devised the experimental set-up and methodology and supervised the research. All the authors contributed substantially to the paper.

Competing interests. There are no competing interests to declare.

Funding. We received no funding for this study.

Acknowledgements. We thank KUYTAM (Koç University Surface Science and Technology Center) and KUTEM (Koç University TÜPRAŞ Energy Center) for their support.

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
