## [Reviewer comments · Royal Society Open Science]

Review History

RSOS-180802.R0 (Original submission)

Review form: Reviewer 1 (Andreas Bräuer)

Is the manuscript scientifically sound in its present form?

Yes

Are the interpretations and conclusions justified by the results?

Yes

Is the language acceptable?

Yes

Is it clear how to access all supporting data?

Yes

Do you have any ethical concerns with this paper?

No

Have you any concerns about statistical analyses in this paper?

No

Recommendation?

Accept with minor revision (please list in comments)

Comments to the Author(s)

The manuscript describes the production of a microreactor with integrated optofluidic waveguide in an silica aerogel and the application of the opto-microfluidic reactor to a photodegradation reaction. The usage of the nanostructured aerogels as cladding in liquid-core waveguides is rather new (presented first in 2017, also by the group of authors) and its application as microphotoreactor is novel. These kind of here presented photoreactors feature some advantages with respect to other microphotoreactors that are also mentioned in the manuscript. In general the contents are clearly presented and follow a very logic order. The figures are of high quality and support the descriptions and statements.

In 2017 the same group of authors published an article in the Journal of Sol-Gel Sci. Technology in which they described/focused on the production of the gel, the supercritical drying of the gel, the treatment of the gel to make it hydrophobic, the generation/drilling of the channels into the gel and the differences in light propagation when the channels are filled with air and with liquid. In this current submission to Royal Society Open Science the authors in the introduction clearly present their new study in the light of optofluidic microreaction and with this clearly distinguish their new work from the previous work. But then in section 2 and section 3 there is a large overlap with the contents already published in 2017. There is no doubt that the contents presented in section 2 and section 3 are essential for the understanding of the manuscript but I recommend to more often refer to the Sol-Gel Sci paper and as a consequence to shorten especially section 2. Section 3 contains a variety of new findings/results that are shown in Figures 5, 7, 8, 9 and 10. These novel results justify the publication of these investigations into the applicability of aerogels as optofluidic microreactors.

Lines 1 to 25 in the section 3.1 read like a repetition of what has already been explained in the section 2. I recommend to clearly focus on the results in the results section and to skip the descriptions regarding the methods.

Figure 5. Can the band between 4000 and 3000 wavenumbers be really assigned to a Si-OH stretching. This way of writing makes the reader anticipate that the stretching might be between Si and OH. I think that this band is due to the stretching between the O and the H. Thus I recommend to label this band not as Si-OH but as O-H. Please also clarify in the text.

Total internal reflection: At several locations within this manuscript the authors mention that the ITR occurs when the light "hits" the channel wall (see for example line 46 in section 3.2 and other lines). To my understanding the TIR occurs when the light reaches the interface between the liquid core (high index of refraction) and the aerogel matrix (lower index of refraction).

In section 3.2 the attenuation of the waveguide is specified with "significantly lower than 3dB/cm". I recommend to provide for comparison the attenuation typical for hollow core photonic crystal fibers.

Review form: Reviewer 2**Is the manuscript scientifically sound in its present form?**

Yes

Are the interpretations and conclusions justified by the results?

No

Is the language acceptable?

Yes

Is it clear how to access all supporting data?

Yes

Do you have any ethical concerns with this paper?

No

Have you any concerns about statistical analyses in this paper?

No

Recommendation?

Major revision is needed (please make suggestions in comments)

Comments to the Author(s)

While this is a very interesting addition to the family of optofluidic waveguides and a potentially useful application, there are some major questions regarding the waveguide performance as listed below:

1. The authors report that the waveguide losses are significantly less than 3 dB/cm but there is no measurement of this loss provided. This is a very important parameter when introducing or characterizing a waveguide so the authors should report it. In this case, measurement of waveguide loss should probably be attempted through a top-down scattering method.
2. The dimensions for the reported waveguides are very large. They are formed with a 2.1 mm drill, which is enormous compared to most optical waveguides. Based on the size of the waveguides, it is unclear just how necessary any "waveguiding" is in this application. A very similar results might be obtained by taking a block of plastic, drilling the same sized hole through it and shining light in the solution. I suggest that such an experiment be done as a comparison to show that there is some utility in the waveguiding nature of the aerogel sample. If the length of the waveguide was very long compares to its width, this utility would be more clear, of course, but in this case where the length to diameter ratio is only around 23, the utility is not so clear.

Decision letter (RSOS-180802.R0)

03-Aug-2018

Dear Dr Erkey,

The editors assigned to your paper ("A New Type of Microphotoreactor with Integrated Optofluidic Waveguide Based on Solid-Air Nanoporous Aerogels") have now received comments from reviewers. We would like you to revise your paper in accordance with the referee and Associate Editor suggestions which can be found below (not including confidential reports to the Editor). Please note this decision does not guarantee eventual acceptance.

Please submit a copy of your revised paper before 26-Aug-2018. Please note that the revision deadline will expire at 00.00am on this date. If we do not hear from you within this time then it will be assumed that the paper has been withdrawn. In exceptional circumstances, extensions may be possible if agreed with the Editorial Office in advance. We do not allow multiple rounds

of revision so we urge you to make every effort to fully address all of the comments at this stage. If deemed necessary by the Editors, your manuscript will be sent back to one or more of the original reviewers for assessment. If the original reviewers are not available, we may invite new reviewers.

- Data accessibility

If you wish to submit your supporting data or code to Dryad (<http://datadryad.org/>), or modify your current submission to dryad, please use the following link:
<http://datadryad.org/submit?journalID=RSOS&manu=RSOS-180802>

- Competing interests

- Authors' contributions

- Acknowledgements

- Funding statement

Please note that Royal Society Open Science charge article processing charges for all new submissions that are accepted for publication. Charges will also apply to papers transferred to Royal Society Open Science from other Royal Society Publishing journals, as well as papers submitted as part of our collaboration with the Royal Society of Chemistry (<http://rsos.royalsocietypublishing.org/chemistry>). If your manuscript is newly submitted and subsequently accepted for publication, you will be asked to pay the article processing charge, unless you request a waiver and this is approved by Royal Society Publishing. You can find out more about the charges at <http://rsos.royalsocietypublishing.org/page/charges>. Should you have any queries, please contact openscience@royalsociety.org.

Kind regards,
Andrew Dunn
Senior Publishing Editor
Royal Society Open Science Editorial Office
Royal Society Open Science
openscience@royalsociety.org

on behalf of Prof R. Kerry Rowe (Subject Editor)
openscience@royalsociety.org

Associate Editor's comments:

Please fully address the concerns raised by the reviewers -- if you do not do so, your paper may be rejected from further consideration.

Comments to Author:

Reviewers' Comments to Author:

Reviewer: 1

Comments to the Author(s)

The manuscript describes the production of a microreactor with integrated optofluidic waveguide in an silica aerogel and the application of the opto-microfluidic reactor to a photodegradation reaction. The usage of the nanostructured aerogels as cladding in liquid-core waveguides is rather new (presented first in 2017, also by the group of authors) and its application as microphotoreactor is novel. These kind of here presented photoreactors feature some advantages with respect to other microphotoreactors that are also mentioned in the manuscript. In general the contents are clearly presented and follow a very logic order. The figures are of high quality and support the descriptions and statements.

In 2017 the same group of authors published an article in the Journal of Sol-Gel Sci. Technology in which they described/focused on the production of the gel, the supercritical drying of the gel, the treatment of the gel to make it hydrophobic, the generation/drilling of the channels into the gel and the differences in light propagation when the channels are filled with air and with liquid. In this current submission to Royal Society Open Science the authors in the introduction clearly present their new study in the light of optofluidic microreaction and with this clearly distinguish their new work from the previous work. But then in section 2 and section 3 there is a large overlap with the contents already published in 2017. There is no doubt that the contents presented in section 2 and section 3 are essential for the understanding of the manuscript but I recommend to more often refer to the Sol-Gel Sci paper and as a consequence to shorten especially section 2. Section 3 contains a variety of new findings/results that are shown in Figures 5, 7, 8, 9 and 10. These novel results justify the publication of these investigations into the applicability of aerogels as optofluidic microreactors.

Lines 1 to 25 in the section 3.1 read like a repetition of what has already been explained in the section 2. I recommend to clearly focus on the results in the results section and to skip the descriptions regarding the methods.

Figure 5. Can the band between 4000 and 3000 wavenumbers be really assigned to a Si-OH stretching. This way of writing makes the reader anticipate that the stretching might be between Si and OH. I think that this band is due to the stretching between the O and the H. Thus I recommend to label this band not as Si-OH but as O-H. Please also clarify in the text.

Total internal reflection: At several locations within this manuscript the authors mention that the TIR occurs when the light "hits" the channel wall (see for example line 46 in section 3.2 and other lines). To my understanding the TIR occurs when the light reaches the interface between the liquid core (high index of refraction) and the aerogel matrix (lower index of refraction).

In section 3.2 the attenuation of the waveguide is specified with "significantly lower than 3dB/cm". I recommend to provide for comparison the attenuation typical for hollow core photonic crystal fibers.

Reviewer: 2

Comments to the Author(s)

While this is a very interesting addition to the family of optofluidic waveguides and a potentially useful application, there are some major questions regarding the waveguide performance as listed below:

1. The authors report that the waveguide losses are significantly less than 3 dB/cm but there is no measurement of this loss provided. This is a very important parameter when introducing or characterizing a waveguide so the authors should report it. In this case, measurement of waveguide loss should probably be attempted through a top-down scattering method.
2. The dimensions for the reported waveguides are very large. They are formed with a 2.1 mm drill, which is enormous compared to most optical waveguides. Based on the size of the waveguides, it is unclear just how necessary any "waveguiding" is in this application. A very similar results might be obtained by taking a block of plastic, drilling the same sized hole through

it and shining light in the solution. I suggest that such an experiment be done as a comparison to show that there is some utility in the waveguiding nature of the aerogel sample. If the length of the waveguide was very long compares to its width, this utility would be more clear, of course, but in this case where the length to diameter ratio is only around 23, the utility is not so clear.

Author's Response to Decision Letter for (RSOS-180802.R0)

See Appendix A.

RSOS-180802.R1 (Revision)

Review form: Reviewer 1 (Andreas Bräuer)

Is the manuscript scientifically sound in its present form?

Yes

Are the interpretations and conclusions justified by the results?

Yes

Is the language acceptable?

Yes

Is it clear how to access all supporting data?

Yes

Do you have any ethical concerns with this paper?

No

Have you any concerns about statistical analyses in this paper?

No

Recommendation?

Accept as is

Comments to the Author(s)

The revised version is recommended for publication.

Review form: Reviewer 2

Is the manuscript scientifically sound in its present form?

Yes

Are the interpretations and conclusions justified by the results?

Yes

Is the language acceptable?

Yes

Is it clear how to access all supporting data?

Yes

Do you have any ethical concerns with this paper?

No

Have you any concerns about statistical analyses in this paper?

No

Recommendation?

Accept as is

Comments to the Author(s)

Authors appear to have satisfied all reviewer concerns.

Decision letter (RSOS-180802.R1)

18-Oct-2018

Dear Dr Erkey,

I am pleased to inform you that your manuscript entitled "A New Type of Microphotoreactor with Integrated Optofluidic Waveguide Based on Solid-Air Nanoporous Aerogels" is now accepted for publication in Royal Society Open Science.

Kind regards,

on behalf of Prof. R. Kerry Rowe (Subject Editor)
openscience@royalsociety.org

Associate Editor Comments to Author:

The authors are commended on the acceptance of their paper.

Reviewer comments to Author:

Reviewer: 2

Comments to the Author(s)

Authors appear to have satisfied all reviewer concerns.

Reviewer: 1

Comments to the Author(s)

The revised version is recommended for publication.

Appendix A

RESPONSE TO REVIEWERS' COMMENTS

Reviewer: 1

The manuscript describes the production of a microreactor with integrated optofluidic waveguide in a silica aerogel and the application of the opto-microfluidic reactor to a photodegradation reaction. The usage of the nanostructured aerogels as cladding in liquid-core waveguides is rather new (presented first in 2017, also by the group of authors) and its application as microphotoreactor is novel. These kind of here presented photoreactors feature some advantages with respect to other microphotoreactors that are also mentioned in the manuscript. In general the contents are clearly presented and follow a very logic order. The figures are of high quality and support the descriptions and statements.

In 2017 the same group of authors published an article in the Journal of Sol-Gel Sci. Technology in which they described/focused on the production of the gel, the supercritical drying of the gel, the treatment of the gel to make it hydrophobic, the generation/drilling of the channels into the gel and the differences in light propagation when the channels are filled with air and with liquid. In this current submission to Royal Society Open Science the authors in the introduction clearly present their new study in the light of optofluidic microreaction and with this clearly distinguish their new work from the previous work. But then in section 2 and section 3 there is a large overlap with the contents already published in 2017. There is no doubt that the contents presented in section 2 and section 3 are essential for the understanding of the manuscript but I recommend to more often refer to the Sol-Gel Sci paper and as a consequence to shorten especially section 2. Section 3 contains a variety of new findings/results that are shown in Figures 5, 7, 8, 9 and 10. These novel results justify the publication of these investigations into the applicability of aerogels as optofluidic microreactors.

Response

The text in Section 2 has been shortened, keeping only the most essential parts in the manuscript. Sub-section 2.3 (Surface Modification with HMDS) has been completely removed from the manuscript; relevant information from this deleted sub-section has been partially added to sub-section 2.2 of the current manuscript.

Lines 1 to 25 in the section 3.1 read like a repetition of what has already been explained in the section 2. I recommend to clearly focus on the results in the results section and to skip the descriptions regarding the methods.

Response

The initial part of Section 3 has been shortened. In the current version of the manuscript, only information necessary for interpretation of our results has been retained.

Figure 5. Can the band between 4000 and 3000 wavenumbers be really assigned to a Si-OH stretching. This way of writing makes the reader anticipate that the stretching might be between Si and OH. I think that this band is due to the stretching between the O and the H. Thus I recommend to label this band not as Si-OH but as O-H. Please also clarify in the text.

Response

The spectral band in question is usually assigned to Si-OH stretching in the literature [1, 2]. In the manuscript, we have adopted the same interpretation and provided appropriate references.

Total internal reflection: At several locations within this manuscript the authors mention that the ITR occurs when the light “hits” the channel wall (see for example line 46 in section 3.2 and other lines). To my understanding the TIR occurs when the light reaches the interface between the liquid core (high index of refraction) and the aerogel matrix (lower index of refraction).

Response

This statement about TIR is correct: when a ray of light is incident upon the interface between the high-index liquid core and low-index aerogel cladding of the waveguide from the liquid side at an angle greater than the critical angle, the light is reflected back to the core liquid. We have changed the phrasing in the manuscript in Section 1 on page 2 to make this clear.

In section 3.2 the attenuation of the waveguide is specified with “significantly lower than 3dB/cm”. I recommend to provide for comparison the attenuation typical for hollow core photonic crystal fibers.

Response

Information on the typical propagation losses of hollow-core photonic crystal fibers has been added to the manuscript in Section 1 on page 2 where these types of reactors are described.

Reviewer: 2

While this is a very interesting addition to the family of optofluidic waveguides and a potentially useful application, there are some major questions regarding the waveguide performance as listed below:

1. The authors report that the waveguide losses are significantly less than 3 dB/cm but there is no measurement of this loss provided. This is a very important parameter when introducing or characterizing a waveguide so the authors should report it. In this case, measurement of waveguide loss should probably be attempted through a top-down scattering method.

Response

Our recent studies cited in the manuscript describe procedures for the measurement of waveguide propagation losses in terms of the input optical power coupled into the waveguide and the output optical power observed after the light has propagated a certain distance z along the waveguide. We have added to the manuscript the measured value of the propagation loss of -1.5 dB/cm reported in [add reference to our paper in the Journal of Sol-Gel Science and Technology] for the same type of optofluidic waveguide.

2. The dimensions for the reported waveguides are very large. They are formed with a 2.1 mm drill, which is enormous compared to most optical waveguides. Based on the size of the waveguides, it is unclear just how necessary any "waveguiding" is in this application. A very similar result might be obtained by taking a block of plastic, drilling the same sized hole through it and shining light in the solution. I suggest that such an experiment be done as a comparison to show that there is some utility in the waveguiding nature of the aerogel sample. If the length of the waveguide was very long compares to its width, this utility would be more clear, of course, but in this case where the length to diameter ratio is only around 23, the utility is not so clear.

Response

For the demonstration of the essential influence of waveguiding on the efficiency of photochemical reactions carried out in optofluidic microphotoreactors, we performed a control experiment, in which we replaced the aerogel-based photoreactor with a piece of plastic tubing (outer diameter 4 mm and inner diameter 2 mm) with L-shaped geometry and the same dimensions as the channel fabricated in the aerogel. Prior to the photolysis experiments, light propagation in the tubing filled with aqueous MB solution was characterized. It was observed that the light could not propagate along the path of the liquid reactant in the liquid-filled tubing, since the contrast of the refractive index between the liquid and the tubing was reversed with respect to that required by TIR. Consequently, a large fraction of the input light coupled

into the tubing was scattered at the tubing elbow. Only a small fraction of the incident light could reach the end of the tubing, exploiting TIR at the interface between the tubing and ambient air. A photograph of the solution-filled tubing with coupled light at 488 nm has been added to the manuscript.

Following the characterization of light propagation in the liquid-filled tubing, photolysis of MB was carried out in the tubing under illumination with activation laser light at 388 nm with an incident light power of 64 mW. At this incident light power, the observed conversion of MB dye in tubing-based photoreactor was only ~10 % after 60 minutes of irradiation, whereas for aerogel-based photoreactor, the observed conversion of MB was 48 % under the same experimental conditions (incident light power and exposure time).

These results that clearly show the importance of the light-guiding effect for the high performance of the microphotoreactor have been added to the manuscript (sub-section 3.3).

REFERENCES

- 1 Loche, D., Malfatti, L., Carboni, D., Alzari, V., Mariani, A., Casula, M. F. 2016 Incorporation of graphene into silica-based aerogels and application for water remediation. *RSC Advances*. **6**, 66516-66523. (10.1039/C6RA09618B)
- 2 Mosquera, M. J., de los Santos, D. M., Rivas, T. 2010 Surfactant-synthesized ormosils with application to stone restoration. *Langmuir : the ACS journal of surfaces and colloids*. **26**, 6737-6745. (10.1021/la9040979)